# Prevalence of Cigarette Smoking among Professionally Active Adult Population in Poland and Its Strong Relationship with Cardiovascular Co-Morbidities-POL-O-CARIA 2021 Study

**DOI:** 10.3390/jcm11144111

**Published:** 2022-07-15

**Authors:** Anna Rulkiewicz, Iwona Pilchowska, Wojciech Lisik, Piotr Pruszczyk, Justyna Domienik-Karłowicz

**Affiliations:** 1LUX MED, Postępu 21C, 02-676 Warsaw, Poland; anna.rulkiewicz@luxmed.pl (A.R.); iwona.pilchowska@luxmed.pl (I.P.); 2Department of Psychology, SWPS University of Social Sciences and Humanities, 03-815 Warsaw, Poland; 3Department of General and Transplantation Surgery, Medical University of Warsaw, 02-014 Warsaw, Poland; wojciech.lisik@wum.edu.pl; 4Department of Internal Medicine and Cardiology, Medical University of Warsaw, 02-005 Warsaw, Poland; piotr.pruszczyk@wum.edu.pl

**Keywords:** cardiovascular diseases, cigarette smoking, professionally active adult population

## Abstract

Smoking is a leading cause of preventable mortality. It affects both the health and economic situation within societies. The aim of the study is to perform an epidemiological analysis of smoking among professionally active adults in Poland in the years 2016–2020 and its Strong Relationship with Cardiovascular Co-morbidities. The article retrospectively analyzed the records of 1,450,455 who underwent occupational medicine examinations between 2016 and 2020. Statistical analyses performed using IBM SPSS Statistics 25 software were performed. In general, irrespective of the year of measurement, 11.6% of women and 17.1% of men declared smoking. After sorting by year of measurement, we found that the percentage of female smokers was decreasing, while that of males remained relatively consistent. In the case of BMI, it was found that among tobacco smokers the percentage of people with normal body weight decreases with successive years of measurement, while the percentage of overweight and level I obesity increases. Moreover, we analyzed in detail the occurrence of particular comorbidities in the group of people who declared smoking. The most common diseases in this group were: arterial hypertension (39%), lipid disorders (26.7%), and hypertension and lipid disorders (16.5%). Active preventive measures are necessary to reduce the number of smokers and the negative impact of smoking on the occurrence of comorbid diseases.

## 1. Introduction

The proportion of cigarette smokers in Europe remains high, with around 21% of adults reporting that they are active smokers [1]. However, cohort studies performed in Europe present the percentage of smokers in the group of 16–20-year-olds as being in decline. This phenomenon is observed in all parts of Europe (Northern, Eastern, and Western Europe) except Southern Europe, where smoking has remained at levels since 1990. The initiation rate in early adolescence (11–15 years) has increased since 1990, especially in Western Europe. The lowest rates of tobacco initiation are observed in Western Europe [2].

In recent years, there has been a decline in the percentage of people who declare themselves smokers. This is attributed to restrictions introduced by individual European countries [3,4]. The Framework Convention on Tobacco Control is a further impetus in the global fight against smoking [5]. However, studies on the age of initiation of smoking are still missing–according to the 2015 Eurobarometer, 19% of Europeans started smoking before the age of 15 [6].

Since 2015, active smoking has been linked to more than five million deaths per year coming from an estimated one billion smokers, while around 600,000 deaths are explained by exposure to passive smoking [7].

Smoking is a leading cause of preventable mortality. It is one of the factors that increase the risk of respiratory diseases, allergies, cardiovascular diseases, and cancer [8]. Young people whose organs are still developing are particularly vulnerable to these diseases. There are many studies that show that exposure to the effects of smoking during the growth period can have a significant impact on health between generations [9,10,11]. In addition, smoking cessation significantly reduces the risk of cancer and heart disease after 12 months of not smoking [12]. Ultimately, people who smoke tobacco products for many years have a lower willingness to quit smoking [13], which results from addiction and low motivation to change their habits [13,14].

Smoking tobacco affects both the health and economic situation within a society. Research by Baker [15] confirmed that tobacco smoking increases absenteeism and decreases professional activity at work among employees from the USA, Europe, and China. These trends improved significantly after cessation of smoking–workers who quit smoking up to four years prior experienced both significant increases in work productivity and fewer days of absence from work. Other studies have shown that US workers who smoke cigarettes lose an average of 2–3 working days per year due to health consequences when compared to workers who have never smoked [16,17,18]. Studies conducted in the Netherlands, Germany, and China gave similar results [19,20,21].

Apart from individual health disorders and occupational troubles, smoking entails very high collateral monetary costs. These are mainly felt as the added costs of providing health care to workers for treatment of diseases resulting from long-term smoking. Still, farther-reaching costs arise from aggregate losses to countries as a result of early smoking mortality [22].

Proper communication between the doctors and patients disclosing their smoking is a very significant factor. Doctors rarely recommend quitting smoking among older adults [23], mainly because the patient is highly addicted or lacks tangible health benefits. However, it is worth noting that quitting smoking in old age may still bring significant health benefits, extend life expectancy and quality [24], and reduce the risk of disability [25]. In addition, quitting smoking can significantly increase the potential benefits for employers, employees, and society as a whole [26].

One of the primary difficulties in developing programs to change the habits of smokers is understanding the more fundamental causes of tobacco addiction; analysis of the ages at which smoking initiation takes place also seems to be important–it should be noted, however, that most publications focus only on the sheer prevalence of smoking in societies. Understanding the reasons underlying tobacco use would almost certainly allow for the development of more effective prevention strategies. Current research indicates that undertaken actions are most effective in lower socioeconomic groups [27].

There have been multiple approaches taken to broadly curb tobacco use. One preventive approach was increasing the price of a pack of cigarettes. Analyses show that this mainly affected young people whose budgets tend to be more sensitive to price increases [28]. Another approach–limiting exposure to tobacco product advertising–was also introduced [29]. The most direct measure–introducing bans on smoking in public places–failed to yield any clear conclusions supporting its efficacy in reducing the percentage of people using tobacco products. European studies conducted in 2019 [30] show that raising prices for tobacco products and limiting places where it is permissible to smoke reduces the number of active smokers mainly in adults up to 65 years of age; the reverse relationship is visible in people over 65 years of age.

Post-quitting productivity gains have prompted many employers to support workers in quitting smoking by investing in tobacco cessation programs and behavioral interventions [31]. Employers incurring the costs of implementing smoking cessation programs also see measurable benefits–the average duration of professional activity of non-smokers is longer than that of active smokers [32].

The aim of the study is to perform a cross-sectional study of smoking among professionally active adults in Poland in the years 2016–2020 and its Strong Relationship with Cardiovascular Co-morbidities.

## 2. Materials and Methods

The article retrospectively analyzed the subsequent records of professionally active adults who underwent occupational medicine examinations between January 2016 and April 2020. In total, the results of 1,450,455 initial, control, and periodic visits as components of occupational medicine certifications were analyzed. During the study, sex, age, height, weight, voivodship of residence, period of validity of medical certification, and data from medical history (subjective assessment of health, smoking) were controlled. We did not exclude any patients. We present data of all subsequent patients. Detailed characteristics of the studied patients are presented in Appendix A.

### Statistical Analysis

Statistical analyses performed using IBM SPSS Statistics 25 software for Windows, Version 27.0. Armonk, NY:IBM Corp were performed [26]. The percentages (with 95% CI) and numbers of observations were used to analyze qualitative data; to characterize the quantitative data: mean (M), standard deviation (SD), median (Me), skewness, kurtosis, and the minimum and maximum statistics were used. Significant statistical results were considered where the probability of making a type I error was less than 5% (*p* < 0.05). For statistical calculations we used: chi-square analysis (Bonferroni’s correction was used to test column proportions) and U Mann–Whitney test.

## 3. Results

The chi-square analysis in the cross tables showed that the percentage of declared smokers slightly decreased with each passing year. It is worth noting, however, that the largest decrease in the percentage of declared smokers occurred between 2016 and other years, taken individually (see Table 1).

### 3.1. Characteristics of Declared Smokers

In general, irrespective of the year of measurement, 11.6% of women and 17.1% of men declared smoking. After sorting by year of measurement, we found that the percentage of female smokers was decreasing, while that of males remained relatively consistent (see Figure 1).

The age of people declaring smoking ranged from 15 to 88 years (M = 37.52; SD = 12.37). There were no considerable changes in declared tobacco smoking in individual age groups in the analyzed years–only in 2020 was there a slight increase in the percentage of smokers in the 35–54 and 55–69 age groups, along with a slight increase in the 18–35 age group (see Table 2).

In the case of BMI, it was found that among tobacco smokers the percentage of people with normal body weight decreases with successive years of measurement, while the percentage of overweight and level I obesity increases (see Table 3).

The average number of months for the occupational medicine certificates was approximately 29 months (M = 29.09; SD = 13.47).

### 3.2. Smoking and Diagnosis According to ICD-10

Table 4 shows the relationship between those who declared cigarette smoking and the occurrence of individual ICD-10 categories (*p* < 0.001). It turned out that in the case of selected categories (such as factors influencing health status and contact with health care and cardiovascular diseases) a higher percentage of diagnoses was associated with people who declared smoking.

Additionally, after dividing cardiovascular diseases into groups, we observed that in the case of ischemic heart disease a higher percentage of cases was found in people who declared smoking; however, for arterial hypertension, the opposite relationship was obtained. The exact results are shown in the Table 5 below.

Analyzing the relationship between selected diseases and declared smoking, it turned out that in the group of smokers a higher percentage of people with hypertension and type 2 diabetes was observed; in the case of lipid disorders, the opposite correlation was obtained (see Table 6).

### 3.3. Cigarette Smoking and Comorbidities

The figure below shows the occurrence of particular comorbidities in the group of people who declared smoking. The most common diseases in this group were: arterial hypertension (39%), lipid disorders (26.7%), and hypertension and lipid disorders (16.5%). The remaining diseases occurred in less than 5% of the patients (see Figure 2).

Moreover, we confirmed that people who declared smoking cigarettes have significantly more diagnosed diseases as compared to people who do not smoke (*p* < 0.001). The obtained results are presented graphically in the Figure 3 below.

## 4. Discussion

In this study, we examined in detail the rates of smoking in Poland using data from 2016–2020. In general, irrespective of the year of measurement, 11.6% of women and 17.1% of men declared smoking. After sorting by year of measurement, we found that the percentage of female smokers was decreasing, while that of males remained relatively consistent. Clearly, the proportion of cigarette smokers in Poland remains high, it is lower than in other European countries [1,2]. In addition, we are very pleased with the delicate downward trend, which, in our opinion, requires intensive legislative changes to strengthen it, i.e., significantly lowering the percentage of active smokers in the group of professionally active Poles [33].

The relationship between smoking and obesity is not clear and published studies have produced conflicting results. Some studies showed no relationship between smoking and obesity, and some give quite different data based on the metabolic effects of nicotine (restricted absorption, reduced calorific intake, increased metabolic rate, and thermogenesis). The Mendelian randomization analysis of UK Biobank data indicated that each standard deviation increment in body mass index (4.6) increased the risk of being a smoker (odds ratio 1.18 (95% confidence interval 1.13 to 1.23), *p* < 0.001) [34,35,36].

In our study, it was clearly found that among tobacco smokers the percentage of people with normal body weight decreases with successive years of measurement, while the percentage of overweight and level I obesity increases. In our opinion, along with the increase in the number of obese patients, it is another factor contributing to the development of comorbidities in this group of patients [34,35].

The relationship between cigarette smoking and the occurrence of individual ICD-10 categories is obviously marked in the group of patients with cardiovascular diseases [37,38,39,40]. It is due to mechanisms, which we present in Figure 4.

Moreover, it turned out that in the group of smokers a higher percentage of people with hypertension, ischemic heart disease, and type 2 diabetes was observed. therefore, in the largest Polish epidemiological study in the field of cigarette smoking, we are consistent with the results of international studies on cardiovascular risk [37,38,39,40]. Moreover, we confirmed that people who declared smoking cigarettes have significantly more diagnosed diseases as compared to people who do not smoke (*p* < 0.001)

## 5. Conclusions

Active preventive actions are necessary to reduce the number of smokers and the negative impact of smoking on the occurrence of comorbid diseases.

## Figures and Tables

**Figure 1 jcm-11-04111-f001:**
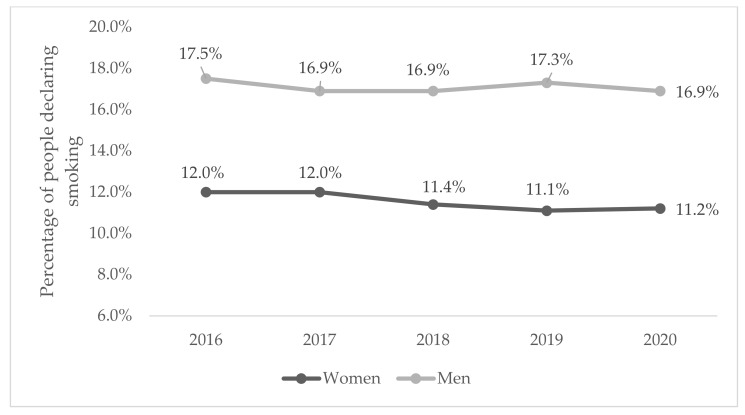
Percentage of declared smokers by sex and year of measurement (95% CI: women group: ±0.3%, men group: ±0.5%).

**Figure 2 jcm-11-04111-f002:**
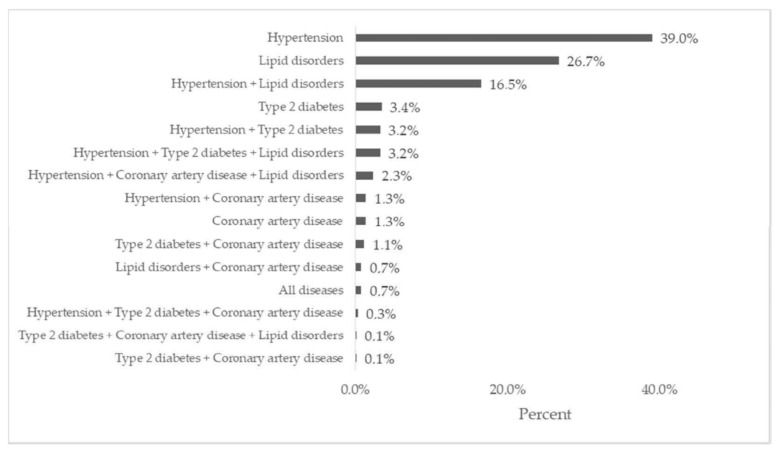
Occurrence of comorbidities in the group of people who declared smoking (95% CI: hypertension: ±0.1%, lipid disorders: ±0.3%; others: ±0.1%).

**Figure 3 jcm-11-04111-f003:**
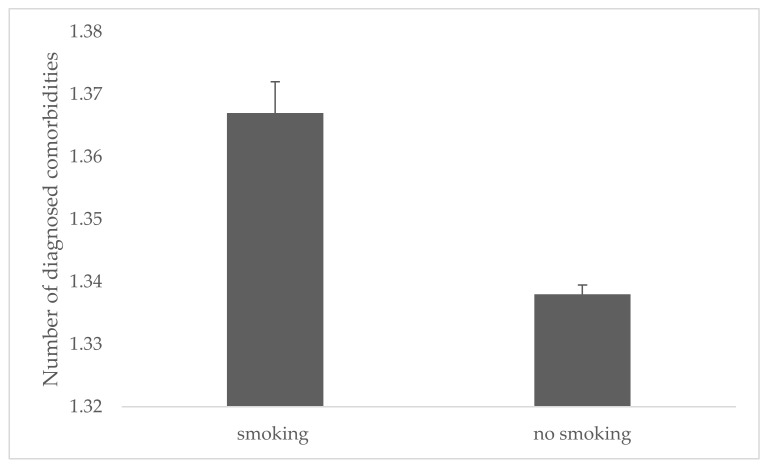
Number of diagnosed comorbidities in the group of people declaring or not smoking (with SD).

**Figure 4 jcm-11-04111-f004:**
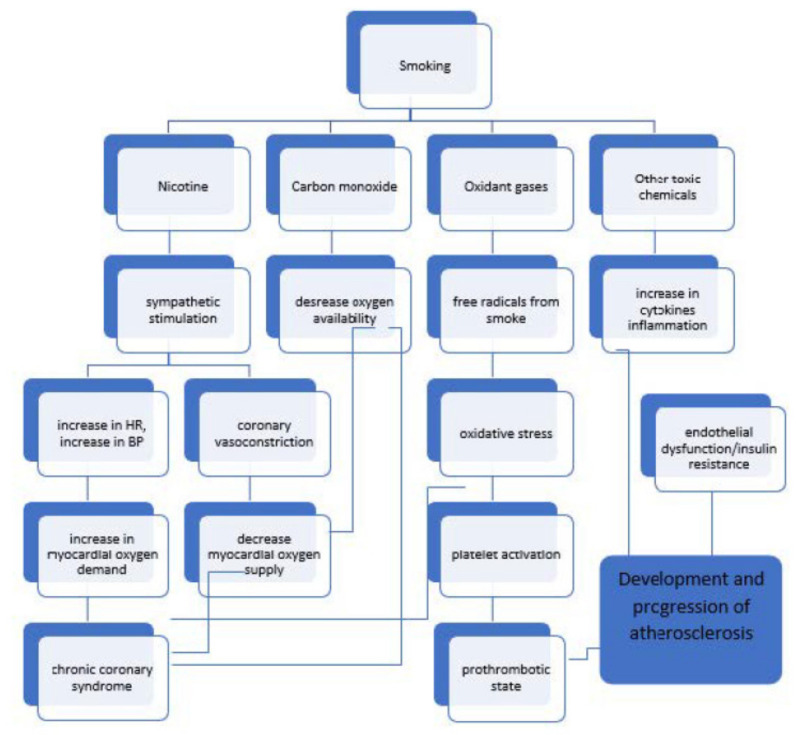
Pathophysiologic mechanisms of tobacco smoke in cardiovascular disease.

**Table 1 jcm-11-04111-t001:** Relationship between smoking and measurement time-data as percentage for the year of measurement (with 95% CI) ^1^.

	2016	2017	2018	2019	2020	Total
No	85.2% _a_ (±0.2%)	85.5% _b_ (±0.1%)	85.8% _c_ (±0.2%)	85.7% _b,c_ (±0.2%)	85.8% _b,c_ (±0.3%)	85.6% (±0.2%)
Yes	14.8% _a_ (±0.2%)	14.5% _b_ (±0.1%)	14.2% _c_ (±0.2%)	14.3% _b,c_ (±0.2%)	14.2% _b,c_ (±0.3%)	14.4% (±0.2%)
Total	100.00%	100.00%	100.00%	100.00%	100.00%	100.00%

^1^ Each letter in subscript represents a subset of the year category whose column proportions do not differ significantly at the level of 5%.

**Table 2 jcm-11-04111-t002:** Relationship between age and measurement time–percentages (with 95% CI) by year of measurement (analysis only for people declaring smoking) ^1^.

	2016	2017	2018	2019	2020	Total
<18	0.1% _a_ (±0.1%)	0.1% _a_ (±0.1%)	0.1% _a_ (±0.1%)	0.0% _a_ (±0.1%)	n/a	0.1% (±0.1%)
18–35	51.1% _a_ (±0.4%)	50.9% _a_ (±0.3%)	51.2% _a_ (±0.3%)	50.8% _a_ (±0.3%)	45.9% _b_ (±0.2%)	50.7% (±0.3%)
35–54	35.9% _a_ (±0.3%)	36.1% _a_ (±0.2%)	35.8% _a_ (±0.3%)	36.3% _a_ (±0.2%)	39.5% _b_ (±0.3%)	36.2% (±0.3%)
55–69	12.8% _a_ (±0.2%)	12.8% _a_ (±0.2%)	12.7% _a_ (±0.2%)	12.6% _a_ (±0.2%)	14.4% _b_ (±0.2%)	12.8% (±0.2%)
>69	0.1% _a_ (±0.1%)	0.1% _a_ (±0.1%)	0.2% _b_ (±0.1%)	0.2% _b_ (±0.1%)	0.2% _b_ (±0.1%)	0.2% (±0.1%)
Total	100.0%	100.0%	100.0%	100.0%	100.0%	100.0%

^1^ Each letter in subscript represents a subset of the year category whose column proportions do not differ significantly at the level of 5%.

**Table 3 jcm-11-04111-t003:** Relationship between BMI and measurement time–percentages (with 95% CI) by the year of measurement (analysis only for people declaring smoking) ^1^.

	2016	2017	2018	2019	2020	Total
Underweight	2.8% _a_ (±0.2%)	2.8% _a_ (±0.2%)	2.7% _a,b_ (±0.3%)	2.8% _a_ (±0.3%)	2.4% _b_ (±0.2%)	2.8% (±0.2%)
Normal body mass	47.0% _a_ (±0.5%)	46.3% _b_ (±0.4%)	45.5% _c_ (±0.4%)	44.8% _d_ (±0.4%)	43.1% _e_ (±0.4%)	45.7% (±0.4%)
overweight	34.2% _a_ (±0.7%)	34.2% _a_ (±0.6%)	34.8% _a,b_ (±0.6%)	34.5% _a,b_ (±0.7%)	35.4% _b_ (±0.7%)	34.5% (±0.6%)
Obesity type I	12.2% _a_ (±0.2%)	12.8% _b_ (±0.3%)	13.1% _b,c_ (±0.3%)	13.4% _c_ (±0.2%)	14.6% _d_ (±0.2%)	13.0% (±0.2%)
Obesity type II	2.9% _a_ (±0.2%)	3.0% _a_ (±0.1%)	3.1% _a_ (±0.2%)	3.4% _b_ (±0.2%)	3.5% _b_ (±0.2%)	3.1% (±0.2%)
Obesity type III	0.8% _a_ (±0.1%)	0.8% _a_ (±0.1%)	0.8% _a_ (±0.1%)	1.0% _b_ (±0.1%)	0.9% _a,b_ (±0.1%)	0.8% (±0.1%)
Total		100.00%	100.00%	100.00%	100.00%	100.00%

^1^ Each letter in subscript represents a subset of the year category whose column proportions do not differ significantly at the level of 5%.

**Table 4 jcm-11-04111-t004:** The relationship between cigarette smoking and the occurrence of individual ICD-10 categories—percentages (with 95% CI) of the smoking category ^1^.

	Smoking		Total
	No	Yes	
Selected infectious and parasitic diseases	0.6% _a_ (±0.05%)	0.5% _b_ (±0.04%)	0.6% (±0.05%)
Cancers	0.6% _a_ (±0.06%)	0.4% _b_ (±0.05%)	0.5% (±0.05%)
Diseases of blood and hematopoietic organs and selected diseases involving immunological mechanisms	0.1% _a_ (±0.01%)	0.1% _b_ (±0.01%)	0.1% (±0.01%)
Disorders of endocrine secretion, nutritional status, and metabolic changes	10.3% _a_ (±0.5%)	8.8% _b_ (±0.4%)	10.0% (±0.4%)
Mental and behavioral disorders	0.3% _a_ (±0.04%)	0.4% _b_ (±0.04%)	0.3% (±0.04%)
Nervous system diseases	0.5% _a_ (±0.07%)	0.5% _a_ (±0.06%)	0.5% (±0.06%)
Diseases of the eye and eye appendages	8.9% _a_ (±0.10%)	9.4% _b_ (±0.09%)	9.0% (±0.10%)
Diseases of the ear and mastoid process	0.9% _a_ (±0.03%)	1.0% _b_ (±0.04%)	0.9% (±0.3%)
Cardiovascular disease	9.0% _a_ (±0.10%)	9.5% _b_ (±0.08%)	9.0% (±0.10%)
Respiratory system diseases	5.4% _a_ (±0.12%)	4.7% _b_ (±0.14%)	5.3% (±0.12%)
Digestive system diseases	2.4% _a_ (±0.15%)	2.1% _b_ (±0.09%)	2.4% (±0.11%)
Diseases of the skin and subcutaneous tissue	2.3% _a_ (±0.22%)	1.7% _b_ (±0.16%)	2.2% (±0.17%)
Diseases of the musculoskeletal system and connective tissue	3.9% _a_ (±0.12%)	3.6% _b_ (±0.17%)	3.9% (±0.13%)
Diseases of the genitourinary system	2.4% _a_ (±0.14%)	1.6% _b_ (±0.14%)	2.3% (±0.14%)
Pregnancy, childbirth and the postpartum period	0.4% _a_ (±0.08%)	0.2% _b_ (±0.06%)	0.4% (±0.06%)
Selected conditions starting in the perinatal period	0.0% _a_ (±0.01%)	0.0% _a_ (±0.02%)	0.0% (±0.01%)
Congenital malformations, distortions, and chromosomal aberrations	0.0% _a_ (±0.01%)	0.0% _b_ (±0.01%)	0.0% (±0.01%)
Symptoms, signs, and abnormal results of clinical and laboratory tests; not elsewhere classified	3.4% _a_ (±0.18%)	2.9% _b_ (±0.16%)	3.4% (±0.16%)
Injury, poisoning, and other specific effects of external factors	2.2% _a_ (±0.03%)	2.1% _b_ (±0.02%)	2.2% (±0.03%)
External causes of illness and death	0.2% _a_ (±0.01%)	0.1% _b_ (±0.02%)	0.2% (±0.01%)
Factors influencing health condition and contact with health services	46.2% _a_ (±0.45%)	50.5% _b_ (±0.41%)	46.8% (±0.42%)
Total	100.0%	100.0%	100.0%

^1^ Each letter in subscript represents a subset of the year category whose column proportions do not differ significantly at the level of 5%.

**Table 5 jcm-11-04111-t005:** Relationship between cigarette smoking and the incidence of individual ICD-10 groups (cardiovascular diseases)–percentages (with 95% CI) for the smoking category ^1^.

	Smoking		Total
	No	Yes	
Acute rheumatic disease	0.0% _a_ (±0.1%)	n/a	0.0% (±0.1%)
Chronic rheumatic heart disease	0.0% _a_ (±0.1%)	0.0% _a_ (±0.1%)	0.0% (±0.1%)
Hypertension	87.0% _a_ (±0.3%)	86.3% _b_ (±0.3%)	86.9% (±0.3%)
Ischemic heart disease	7.1% _a_ (±0.2%)	8.8% _b_ (±0.1%)	7.4% (±0.2%)
Cardiopulmonary syndrome and pulmonary circulation diseases	0.0% _a_ (±0.1%)	0.0% _a_ (±0.1%)	0.0% (±0.1%)
Other heart conditions	1.8% _a_ (±0.1%)	1.8% _a_ (±0.1%)	1.8% (±0.1%)
Cerebral vessel diseases	0.2% _a_ (±0.1%)	0.3% _a_ (±0.1%)	0.2% (±0.1%)
Diseases of arteries, arterioles, and capillaries	0.4% _a_ (±0.1%)	0.4% _a_ (±0.1%)	0.4% (±0.1%)
Diseases of the veins, lymph vessels, and lymph nodes, not elsewhere classified	3.3% _a_ (±0.3%)	2.3% _b_ (±0.2%)	3.1% (±0.2%)
Other and unspecified disorders of the circulatory system	0.1% _a_ (±0.1%)	0.1% _a_ (±0.1%)	0.1% (±0.1%)
Total	100.0%	100.0%	100.0%

^1^ Each letter in subscript represents a subset of the year category whose column proportions do not differ significantly at the level of 5%.

**Table 6 jcm-11-04111-t006:** Relationship between cigarette smoking and the occurrence of selected diseases–percentages (with 95% CI) of the smoking category ^1^.

	Smoking		Total
	No	Yes	
Hypertension	44.5% _a_ (±0.1%)	48.7% _b_ (±0.1%)	45.1% (±0.1%)
Type 2 diabetes	8.2% _a_ (±0.2%)	8.9% _b_ (±0.2%)	8.3% (±0.2%)
Lipid disorders	43.6% _a_ (±0.4%)	37.5% _b_ (±0.3%)	42.8% (±0.3%)
Coronary artery disease	3.6% _a_ (±0.1%)	5.0% _b_ (±0.1%)	3.8% (±0.1%)
Total	100.0%	100.0%	100.0%

^1^ Each letter in subscript represents a subset of the year category whose column proportions do not differ significantly at the level of 5%.

## Data Availability

Not applicable.

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
