# Peer review of "Prevalence of Cigarette Smoking among Professionally Active Adult Population in Poland and Its Strong Relationship with Cardiovascular Co-Morbidities-POL-O-CARIA 2021 Study"

_jcm, 2022, doi:10.3390/jcm11144111_

Round 1

Reviewer 1 Report

In this study, the aim of the study is to perform an epidemiological analysis of smoking among professionally active adults in Poland in the years 2016-2020 and its strong relationship with cardiovascular co-morbidities.  

This study was not with high novelty.

Minor Concerns

1.  In the section of 1.1, whether the last sentence was not well written, please double-check it.

2.  In the section of Discussion, the potential mechanism by which smoking has bad effects on cardiovascular system should be mentioned.

Author Response

Point 1

In this study, the aim of the study is to perform an epidemiological analysis of smoking among professionally active adults in Poland in the years 2016-2020 and its strong relationship with cardiovascular co-morbidities.  

This study was not with high novelty.

Response 1: First of all, we thank you very much for your comments that have helped us improve our publication.Thank you for your comment 1. We agree that many countries ( governement/epidemiologists/cardiologists/pulmonologists) provide this kind of data. But in Poland we do not share these kind of data. What is more in the very important group of professionaly active/employees we don’t know such unique data and their relations.

Point 2

In the section of 1.1, whether the last sentence was not well written, please double-check it.

Thank you for your comment 2. We improved this sentensce according to your

Point 3

In the section of Discussion, the potential mechanism by which smoking has bad effects on cardiovascular system should be mentioned.

Thank you for your comment 3. We improved this part according to your remark

Reviewer 2 Report

I would like to congratulate the authors for conducting the present research.

Here goes a few of my concerns:

The title and Abstract looks acceptable.

I suggest the authors to place the keywords by alphabetic order.

I recommend the authors to remove the subheading from the Introduction. Traditionally the Introduction is a single text with the goals to introduce a certain topic while leading the reader to the study rationale an lately to the aim sentence. The split is subheading cut the narrative and makes difficult to correctly conduct the required process.

I suggest the authors to be more precise in the aim sentence regarding the study design. Therefore I suggest the change from epidemiological analysis to cross-sectional study (which is the present study design, and also a form of epidemiological assessment).

I recommend to remove the “secondary goal” from the aim sentence. This might be a secondary goals that should be debate in the Discussion, but should be not mentioned as a study aim since nothing in the methods directly was conducted to answer that question… it is just theoretical debate in relation to the results.

The Material and Methods are by far not acceptable. The author give go clue of how the data was assessed, how it was assessed, who assessed it, when it was assessed, inclusion/exclusion criteria, exclusion rate, exclusion reasons, potential sources of bias, so on, so on. I recommend the authors to pick a Cross-sectional study guide or check list and accomplish it (for instances STROBE). The present methodology is not a scientific methodology by any means.

The result (proportions) should be presented with a 95% confidence interval.

Once again the Discussion has severe gaps. The authors forgot to debate the study strength, study limitations, generalization of the results, internal and external validity and future studies perspectives.

The reference list are not in accordance to the journal instructions

Author Response

Response to reviewer 2 comments

Point 1

I would like to congratulate the authors for conducting the present research.

Response 1: First of all, we thank you very much for your comments that have helped us improve our publication.

Point 2

Here goes a few of my concerns:

The title and Abstract looks acceptable.

Thank you

 Point 3

I suggest the authors to place the keywords by alphabetic order.

Thank you for your comment. We did it

Point 4

I recommend the authors to remove the subheading from the Introduction. Traditionally the Introduction is a single text with the goals to introduce a certain topic while leading the reader to the study rationale an lately to the aim sentence. The split is subheading cut the narrative and makes difficult to correctly conduct the required process.

 Thank you for your comment. We did it

Point 5

I suggest the authors to be more precise in the aim sentence regarding the study design. Therefore I suggest the change from epidemiological analysis to cross-sectional study (which is the present study design, and also a form of epidemiological assessment).

  Thank you for your comment. We introduced a change according to your suggestion

Point 6

I recommend to remove the “secondary goal” from the aim sentence. This might be a secondary goals that should be debate in the Discussion, but should be not mentioned as a study aim since nothing in the methods directly was conducted to answer that question… it is just theoretical debate in relation to the results.

Thank you for your comment. We introduced a change according to your suggestion

 Point 7

The result (proportions) should be presented with a 95% confidence interval.

Thank you for your comment. We introduced many changes according to your suggestion

 Point 8

Once again the Discussion has severe gaps. The authors forgot to debate the study strength, study limitations, generalization of the results, internal and external validity and future studies perspectives.

Thank you for your comment. We introduced a change according to your suggestion

 Point 9

The reference list are not in accordance to the journal instructions

Thank you. We corrected it.

Round 2

Reviewer 1 Report

No

Reviewer 2 Report

Dear author. I have no more concerns.